# Savings Groups for Social Health Protection: A Social Resilience Study in Rural Tanzania

**DOI:** 10.3390/diseases10030063

**Published:** 2022-09-09

**Authors:** Brigit Obrist, Angel Dillip, Albino Kalolo, Iddy M. Mayumana, Melina Rutishauser, Vendelin T. Simon

**Affiliations:** 1Department of Social Sciences, University of Basel, 4051 Basel, Switzerland; 2Apotheker Consultancy (T) Limited, Health Access Initiative, Dar es Salaam P.O. Box 70022, Tanzania; 3Department of Public Health, St Francis University College of Health and Allied Sciences, Ifakara P.O. Box 175, Tanzania; 4Kilombero Valley Health and Livelihood Promotion, Ifakara P.O. Box 43, Tanzania; 5Anthropology Unit, University of Dar es Salaam, Dar es Salaam P.O. Box 35091, Tanzania

**Keywords:** savings groups, social health protection, social resilience, participatory qualitative research

## Abstract

Global health experts use a health system perspective for research on social health protection. This article argues for a complementary actor perspective, informed by the social resilience framework. It presents a Saving4Health initiative with women groups in rural Tanzania. The participatory qualitative research design yielded new insights into the lived experience of social health protection. The study shows how participation in saving groups increased women’s collective and individual capacities to access, combine and transform five capitals. The groups offered a mechanism to save for the annual insurance premium and to obtain health loans for costs not covered by insurance (economic capital). The groups organized around aspirations of mutual support and protection, fostered social responsibility and widened women’s interaction arena to peers, government and NGO representatives (social capital). The groups expanded women’s horizon by exposing them to new ways of managing financial health risk (cultural capital). The groups strengthened women’s social recognition in their family, community and beyond and enabled them to initiate transformative change through advocacy for health insurance (symbolic capital). Savings groups shape the evolving field of social health protection in interaction with governmental and other powerful actors and have further potential for mobilization and transformative change.

## 1. Introduction

Conversations with Marcel Tanner sparked our interest in the research topic of this study. It all began with the Dar es Salaam Urban Health Project in Tanzania (1990–2002) and the engagement of critical medical anthropology in urban health research [1,2]. The obvious interlinkages between health and livelihood first drew our attention to the advantages of combining the epidemiological risk perspective with a vulnerability perspective. The risk perspective is population-centered and highly relevant for determining “the probability of an adverse outcome, or a factor that raises this probability” [3] (p. xiii). The vulnerability perspective is people-centered and directs attention to people’s exposure to real or potential harm while lacking the means to cope or to protect themselves [4]. 

In the course of further research, we became more critical of the vulnerability perspective because it emphasizes exposure and lack of means, conceptualizing human beings primarily as victims. Ethnographic studies in a low-income neighborhood of Dar es Salaam shifted the interest to the concept of resilience, which considers people as actors rather than victims. The new findings from Dar es Salaam clearly showed that comparable urban health risks did not affect women equally, largely because they created and made differential use of response options within the household but also by reaching out to social and economic groups, institutions and organizations [5].

Within the Swiss National Centre of Competence in Research North-South (2001–2013), Marcel Tanner opened a space for expanding the conversation about risk, vulnerability and resilience to urban and nomad health research in West Africa [6,7]. He also encouraged an interdisciplinary research group from Europe, Africa, Asia and South America to explore the potential and limitations of the resilience concept in sustainable development research [8]. Linking strands of resilience thinking from ecology, psychology and development studies with social and cultural theory, we developed a new framework for studying “multi-layered social resilience” [9]. The current study draws on the social resilience framework to gain new insights on social health protection.

Global health experts commonly frame research on social health protection in terms of universal health coverage (UHC). The goal of UHC encompasses three interrelated components: (1) the full spectrum of health services according to need; (2) financial protection from direct payment for health services when consumed; and (3) coverage for the entire population [10]. Since fragmentation of health financing is a common challenge in low- and middle-income countries, health economists argue for a health system perspective. The crucial question for them is how a given scheme or financing mechanism influences progress towards UHC at the population level [11]. The Swiss TPH, for instance, initiated the Health Promotion and System Strengthening Project (HPSS) to support the Tanzanian government with an array of interventions to move towards UHC [12]. One of them is the re-design of the Community Health Fund (CHF) into a voluntary health insurance for people living in the informal sector and rural areas [13]. The national rollout of the “improved CHF” (iCHF, in Kiswahili: CHF iliyoboreshwa) began in May 2018. An annual premium of Tanzania Shilling (TZS) 30000 (about USD 13) entitles a household of up to six persons to access all medical services in accordance with the referral procedure at all levels of government service centers. However, a number of governance issues act as barriers to the implementation of the iCHF [14], and the active iCHF membership rate (2.84% by the end of April 2021) remains far below expectations [15]. As outlined in Tanzania’s Health Sector Strategic Plan V 2021–2026 “Leaving No One Behind”, the government plans to “mobilize citizens to join health insurance schemes to ensure that every citizen has access to health care without financial constraints” [16] (p. xvi). 

This article calls for an expansion of the conventional understanding of social health protection and presents a novel actor perspective to complement the system perspective Several studies have shown that actors living in contexts of livelihood insecurity tend to raise financial resources for health care from various social networks, including family, relatives, friends and neighbors [17]. They organize themselves, for instance in Rotating Credit and Saving Associations (ROSCAS) [18] and try to link up with diverse supportive arrangements such as micro-insurance, community funds, mutual health organizations, rural health insurance, revolving drug funds and church-based groups [19]. 

From an actor perspective, savings groups are of particular interest. Up to now, they have been neglected in the social health protection literature. Economists working in informal settings emphasize that joining a health insurance is more cost-effective than saving because of pooling resources and risk [20]. For millions of people around the world, though, ROSCAs and other forms of savings’ groups are highly attractive. They emphasize sociability rather than economic effectiveness [18]. ROSCAs are popular among the poor and in fluctuating economic conditions but they also thrive in more affluent social strata in Africa and Asia [18]. Over the past 30 years, hundreds of international and local NGOs have mobilized people to participate in savings groups, at least partly in response to drastic Social Adjustment Programs introducing user fees in education and health care. A network of practitioners formalized some key principles of ROSCAS and developed the Village Saving and Lending Association (VSLA) methodology that has spread to 77 countries with over 20 million active participants worldwide [21]. In Tanzania, the international NGO CARE introduced VSLA in Zanzibar in 1998 and since then, together with many other agencies, facilitated the establishment of several thousand VSLA groups [22]. The 2017 FinScope Survey Tanzania found that 16% (4.4 million) of Tanzanians aged 16 and older were members of savings groups [23], and CARE Tanzania [22] estimated that they generated collective potential savings of nearly 400 billion Tanzanian Shillings (TZS) (about USD 174,700). The figures would probably double or triple, if all self-help groups and microcredit groups with similar saving and lending mechanisms would be included.

Although there are many variations, classic savings groups share common principles [24]. A group of 20–30 people meet regularly to save together and take small loans from those savings. The activities of the group run in cycles, commonly of one year. At the end of a cycle, the management committee distributes the accumulated savings and the loan profits back to the members. Further characteristics are that members elect their own management committee and decide who can join the group and who has to leave it. The committee keeps the cash and records in a lockable cash box. Groups develop their rules regulating their meeting schedule, how much can be saved at each meeting, loan conditions and interest rates, and how the savings and interest paid on loans are shared out at the end of the cycle. Most groups choose to contribute to a social fund for covering the costs of small emergencies [24]. 

In terms of programming, the VSLA methodology differs significantly from other approaches [24]. Microcredit programs are funded and implemented by external agencies, including global development agencies, for-profit or non-profit microfinance institutions, NGOs and government agencies. Promotors of savings groups argue for a de-institutionalization of financial services for the poor. Based on evidence from experimentation and research, they emphasize that poor people can make savings to meet small credit needs (USD 5–500) without external finance and costly external management [24] (p. 10). VSLA facilitating agencies organize poor people into self-elected savings groups, deliver training and supervision at a low cost and do not generate any revenue from the groups. 

Our study links these different strands of research and action by exploring real and potential contributions of savings groups to social health protection. Framing our study in terms of social resilience, we are particularly interested in exploring how women savings groups strengthen capacities to not only cope with financial risks in health, but to search for and create options for protection against financial health risks. The aim is to study actual saving and lending activities, but also the forming of more intangible capacities. The savings group literature often refers to non-financial benefits as “social capital” [24]. We suggest a refined analysis and will be especially attentive to how capacities to access social, cultural and symbolic capital, for instance the capacity to aspire, are formed “in the thick of social life” by lending vision and horizon to immediate concerns [25]. We thus intend to show that a resilience approach to social health protection generated by and through savings groups helps to explore and identify potentials for mobilization and transformative change in social health protection, including a move towards the global aspiration of UHC.

## 2. Materials and Methods

### 2.1. Study Design and Settings

This study used a participatory qualitative research design to gain new insights into local experiences and aspiration of social health protection by and through women groups. A hallmark of participatory qualitative research is that “it enables the lived experience of participants to be integrated with theoretical and academic knowledge” and thus to the generation of new knowledge and insights [26] (p. 5). Research participants may include academic researchers, professionals in the fields of health care, education and social welfare, policy makers, and members of diverse groups and communities whose life or work is the subject of the research [27]. In this study, academic researchers from Switzerland and Tanzania collaborated with professionals of the Tanzanian NGO Kilombero Valley Health and Livelihood Promotion (KV-HELP) located in the rural town Ifakara, and with a network of KV-HELP supported women groups spread out across the Kilombero Valley.

Situated in southern Tanzania, the Kilombero Valley is a vast floodplain delimited by the Udzungwa Mountains in the north and the Mahenge Mountains in the south. Most inhabitants are farmers, taking advantage of the flooding to cultivate rice, together with maize, the main staple food and most important cash crop in the area. In administrative terms, the valley forms part of the Morogoro Region, with a population of more than 2 million at the last census in 2012 [28], and was divided into the three districts of Kilombero, Malinyi and Ulanga at the time of this study.

### 2.2. Background Information on KV-HELP and the Women Groups

KV-HELP has grown out of the ACCESS Project, implemented by the Ifakara Health Institute from 2003–2012 with financial support from the Novartis Foundation for Sustainable Development (NFSD) and technical support from the Swiss Tropical and Public Health Institute (Swiss TPH) [29]. Building on the experience of the overall project [30] and on the findings of a preliminary study [31], the ACCESS team designed a small intervention aimed at strengthening women’s capacity to build resilience towards health-related financial shocks. From 2008 onwards, the ACCESS Project supported 10 women groups, with a grant of 2–3 million Tanzanian Schillings (TZS, about USD 860–1290) per group to start a revolving fund and with basic training in income generation and health promotion. In 2010, the project encouraged the women groups to set up an additional Social Fund in order to save for bigger expenses in education, health care and funerals. In addition, the project team provided information and education sessions on the CHF and trained women groups in promoting the scheme in their communities. In 2012, the ACCESS team set up the NGO KV-HELP to take over. KV-HELP added 7 new women groups to the network, also with the help of private sponsors such as Marcel Tanner and Claus Leisinger (then Director of NFSD). 

### 2.3. The Research Process

In 2017, the research team reviewed past activities at the desk and conducted an ethnographic assessment of savings groups in all 17 villages from 1 April to 15 July. Based on the insights from the review and the assessment, the academic and professional team members wrote a proposal for a small Saving4Health initiative to complement ongoing group activities. This initiative had a double aim: (1) to boost health protection related activities already introduced by the ACCESS Project, and (2) to critically engage with the pilot rollout of the improved CHF scheme in the study area, which began in 2016. The team raised funds and involved all women groups in the detailed planning during visits in June/July 2018. 

The idea was to embed the Saving4Health initiative in the local economy and start the first saving cycle at the end of the rice cultivation season. The women would thus be able to gradually build up capital by selling surplus rice and by engaging in small income generating activities until December. The groups would then have sufficient savings to capitalize two loan funds from which the members could borrow for the next rice cultivation season and for managing livelihood risks including health care costs (Savings Fund) as well as small emergencies linked to health, education or funerals (Social Fund). 

In July 2018, the research team organized a kick-off workshop with local authorities in Ifakara. The workshop opened a space for the finalization of the Saving4Health guideline to facilitate the process of collecting and recording women’s savings and loans during regular group meetings. The Ifakara team provided each women group with a saving kit (i.e., a metal cash box with padlocks, two plastic containers, passbooks for each member, a stamp and a stamp pad for entries into the passbook, and a small calculator). As an incentive, KV-HELP offered a matching grant of TZS 640000 (about USD 280) for each women group that saved this amount or more in the first cycle of 12 months. 

In the first saving cycle (July 2018 to June 2019) and the second saving cycle (July 2019 to June 2020), the research team reviewed progress during its quarterly visits in the villages. At these meetings, the professional and academic team members engaged the women, and if needed other stakeholders, in moderated discussions and individual interviews about social health protection to review experiences and aspirations, clarify questions and search for solutions of emerging problems. 

To complement the view gained during the review meetings with all women savings groups, the research team invited a Tanzanian (Mickness Mshana)—Swiss (Anja Orschulko) tandem of MA students in African Studies from the University of Basel to conduct an in-depth study of the Neema women group in Igota. The team considered this group as a positive example for social health protection and asked the students to take a critical look at claims about women’s empowerment (M.M.) and the links between the savings group and the iCHF (A.O.). The students lived in Igota village for six weeks in June and July 2019 and participated in weekly group meetings as well as meetings of the research team in Igota and Ifakara. 

The research languages were Kiswahili (the widely spoken and official language in Tanzania) and English, depending on the context and language competencies of those present. During the women group visits, the Ifakara team transferred group records into a digital monitoring tool and took notes of group discussions, observations and interviews. The MA students’ audio-recorded the interviews in Kiswahili and transcribed and translated them into English. The academics and professionals shared their analysis and interpretation with the women groups in an ongoing discussion, expanding and refining their understanding of local experiences and aspirations of social health protection. The academic team members integrated these insights with theoretical knowledge. In the presentation of our findings, we use only the village names to refer to the groups for reasons of confidentiality and readability. The only exceptions are the Neema group members of Igota who insisted on being mentioned by name.

### 2.4. Conceptual Framework

Studying social health protection from a resilience perspective is in line with the past work of the ACCESS Project and KV-HELP and, at the same time, opens up new lines of inquiry. Social resilience has been defined as “the capacity of actors to access capitals in order to—not only cope with and adjust to adverse conditions (that is, reactive capacity)—but also search for and create options (that is, proactive capacity) and thus develop increased competence (that is, positive outcomes) in dealing with a threat” [9] (p. 289). Guided by this definition, our study distinguishes between proactive and reactive capacities in building social resilience to financial health risks and thus increase social health protection. 

From the social resilience perspective used here, the experiences of and aspirations for social health protection are not just about proactive and reactive capacities to access economic capital (mainly cash or assets). As studies of social inequality have shown [32], it is just as important to consider capacities to access social capital (various kinds of valued relations) and cultural capital (knowledge, skills and education) and the ways in which all three forms of capitals can be mobilized, combined and transformed to increase symbolic capital (social recognition, honor, reputation). Our study will thus explore how savings groups foster and develop pro- and reactive capacities to access economic, social, cultural and symbolic capitals, alone and in combination to improve women’s competence in dealing with financial health risks. 

The framework used here further emphasizes that social resilience is multi-layered, built through enabling interactions of actors on different layers of society. Ideally, enabling interactions have a cascading effect and reinforce each other. The study will thus explore enabling interactions that foster—or at least have the potential to foster—the building of resilience for social health protection. 

## 3. Results

Guided by the social resilience perspective, we structure the presentation of our key findings by capacities to access capitals, moving from economic capital to symbolic capital. 

### 3.1. Capacities to Access Economic Capital

Saving is a proactive financial capacity as it opens an opportunity to access economic capital in the future. By opening a minimally structured space for financial group action and with a small financial incentive after the first year, the Saving4Health initiative fostered women to develop their saving capacity collectively and individually. Since this is a qualitative study and since we want to protect confidentiality regarding the financial situation of the groups, we only report on the median and the range of saving values. 

Since the Saving4Health initiative follows the basic VSLA model, the women groups established a Savings Fund for flexible livelihood purposes (from agriculture to business, education and health care) and a Social Fund providing protection against small emergencies (including illness-related costs and funerals). During both cycles and in all groups, women saved more money in their Savings Fund than in their Social Fund. 

We first consider the Savings Funds. At the end of the first cycle, the median of the total group assets was USD 1220 (in a range of USD 225–5515), at the end of the second cycle USD 1378 (in a range of USD 586–5858). These findings show that within a saving cycle of 12 months, more than half of the women groups generated Saving Fund assets that were comparable to the original ACCESS grants (USD 860–1290). We had lower expectations when we defined the cut off point for the matching fund (USD 280). With one exception, the women groups saved more than USD 280 and received the matching fund. The difference between the highest and the lowest group assets per cycle was striking. We also found marked differences in individual saving capacity. At the end of the first cycle, individual savings in the group with the lowest asset (USD 225) were in a range of USD 1–16, in the group with the highest asset (USD 5515) in a range of USD 41–341. At the end of the second cycle, individual savings in the group with the lowest asset (USD 586) ranged between USD 0–70, in the group with the highest total asset (USD 5858) between USD 0–327. Reasons for this variation will be discussed below.

In the Social Funds, the median of the total group assets was USD 176 (in a range of USD 53–1355) at the end of the first cycle and USD 184 (in a range of USD 44–1508) at the end of the second cycle. In addition, in terms of contributions to the Social Fund, individual financial capacities differed across and within groups. At the end of the first cycle, individual savings in the group with the lowest asset (USD 53) were in a range of USD 0.3–3.9, in the group with the highest asset (USD 1355) in a range of USD 7–45. At the end of the second cycle, individual savings in the group with the lowest asset (USD 44) ranged between USD 0.5–1.6, in the group with the highest total asset (USD 1508) between USD 37–84.

In group discussions and interviews, explanations of this variation centered around three broad categories: (1) Contextual factors (i.e., geographical location, supportive governmental structures, business opportunities, means of transport, access to markets); (2) skills in financial management (e.g., basic bookkeeping and accounting, group rules, for instance regarding the frequency and amount of regular savings); and (3) group dynamics (e.g., mobility, conflicts, illness, death). Two groups, for instance, accumulated very low total assets during the first cycle and realized that their skills were too limited. They asked extension officers for help in the second cycle and were able to greatly increase their Saving Fund asset. In the most successful groups, leaders advised new members to start slowly. Most groups allowed members to pause or contribute only small amounts now and then if they or a family member fell ill, faced a major conflict or had other serious problems, including weakness due to old age. As was to be expected, a varied and changing mix of contexts and conditions in which savings groups operated enabled and constrained women’s pro-active capacities to save and generate financial options for reducing economic risks, including health care related costs, in the future.

From a social resilience perspective, actually taking a loan improves the reactive capacity to master a health-related financial risk. In a random sample interviewed during the first cycle, we found that women from eight out of 17 groups had used loans from the Savings Fund and/or the Social Fund to pay for health care expenses. The values of these loans were in the range of USD 4–131 with a median of USD 33. 

An important issue raised by women in several group discussions concerned interest rates. Some women pointed out that taking a loan from the Savings Fund for health-related reasons does not make sense economically. Most groups charged an interest of 5–10 percent on Saving Fund loans, which was reasonable for financing productive activities. The Social Fund, on the other hand, was meant for financing emergencies including health care costs. Most groups charged only 2 percent interest on these loans; other groups allowed members to take interest-free loans. Other members pointed out that only the Savings Fund could provide higher loans for bigger problems. Still other argued that if they needed the money quickly, they simply spent whatever they had left from loans for other purposes. The professional member of the research team suggested increasing contributions to the Social Fund. This would make sense for covering emergency health care costs and even the iCHF premium. Whatever decisions groups and individual members took after these discussions, the exchange of ideas itself had a value for many women. As some of them pointed out, it was through such discussions that even silent members gradually expanded their understanding of health care financing. We will elaborate this point in the following sections.

At the end of the second cycle (see Table 1), we found that more than half of the women who had taken Social Fund loans used them for health care expenses (88 out of 152, 58 percent). This relation was much smaller for women using Savings Fund loans (22 out of 284, 8 percent) and ACCESS loans (6 out of 234, 3 percent). In some of these groups, the above-mentioned discussion had taken place while we were there. Whether the group discussions had led to a change in group and/or individual action is difficult to say based on our data. Once again, we found that differences of loan values between groups were remarkable, with a range between USD 4–594 and a median of USD 22. 

The most striking finding on loans was that group rules allowed a flexible use of loans for health care costs. Members could contribute to direct or indirect costs for themselves or for a sick family member or pay for travel expenses for taking care of a sick family member in another village or in a town. They would also take loans to pay for tests and medication not covered by the iCHF or for transport to acquire iCHF or National Health Insurance Fund (NHIF) covered treatment in a referral health facility. Still others used loans to pay for medicines in private retail drug shops or for services in private or faith-based health facilities.

In a social resilience perspective, joining a health insurance can also be seen as a capacity to access economic capital to protect against financial health risk (proactive capacity). Already at the end of the first cycle, a number of women in the random sample claimed to be iCHF members, but it was difficult to verify their reported membership. The in-depth study of the Neema group in Igota found that all women had decided to collectively enroll in the new CHF in July 2018. 

At the end of the second cycle, we were able to check women’s iCHF membership cards (see Table 2). The most striking finding here was that a remarkable 33 percent of the women (152 out of 458) were active iCHF members. In five groups including Neema, nearly all women had joined, and we will look at some of the reasons in the next sections. In most groups, women mentioned to have used savings from the Social Fund to pay the annual iCHF premium of TZS 30000 (about USD 13), as the Saving4Health guideline suggested. Some women (8 percent) were active members of the NHIF, either through their employment as nurse, teacher or prison officer or through a formally employed spouse or relative. A few savings group members (2 percent) were exempted by policy (older than 60 years) and/or waived as Tanzania Social Action Fund (TASAF) recipients (i.e., identified as poor by the Local Government Authorities). 

The moderated group discussions during the visits of the research team, but also the regular group meetings, opened a space for an exchange of experiences and aspirations concerning these diverse health protection options. Women shared positive and negative experiences with the NHIF and the iCHF, for instance receiving good diabetic services through NHIF membership but facing many problems to acquire even basic care through iCHF membership. One woman reported that she almost did not renew the iCHF membership because the cost had been higher than the benefit, but then she had to acquire a surgery in the second year, which the iCHF covered because she went through the referral system of the public health system. The distribution of TASAF waivers was often a topic of hot debates within the communities, as members contested the entitlement of others. The professional member of the research team could clarify some misunderstandings concerning TASAF regulations. 

Setting up an opportunity for saving does not automatically translate into improved group and individual capacities to access economic capital. Saving groups foster and develop additional capacities for transformative change.

### 3.2. Capacities to Access Social Capital

The Saving4Health initiative could draw on social relations that women had built up, maintained and cultivated over a long time. Most groups of the KV-HELP network formed in the early years of the new millennium and stayed together until now. In a situation of unstable marriages due to illness, death and conflict, the groups generated a sense of belonging, safety and trust. Women referred to their group as kikundi mama (the group you can always turn to if you face problems), as substituting the husband or as an asset. Rather than having a designated Social Fund, some groups had a rule to contribute a certain amount whenever one of the members faced an emergency. Sometimes members decided to come together to enjoy playing merry-go-round (ROSCAS) for a few months. 

We further found that savings groups had mobilized women around the locally familiar aspiration of mutual support to improve livelihood and master future emergencies including illness. Women in 11 groups had organized themselves as ROSCAS or around joint economic activities (e.g., cultivating a rice field), others around a grant (from the Participatory Agricultural Development and Empowerment Project), a specific purpose (helping People Living with AIDS) or to acquire training in VSLA (through CARE Wezesha). When the ACCESS Project/KV-HELP invited these groups to join their program, and again in the planning of the Saving4Health intervention, the groups went through a process of aligning existing and new values and organizational forms. We consider this social flexibility as another hallmark of savings groups, or perhaps more precisely of saving and credit oriented self-help groups. We continue to refer to the women groups in our study as savings groups for reasons of readability but also because our initiative emphasized the saving component. 

The savings groups allowed women to set rules that match their livelihoods, to adapt them to their needs and to control how the money was collected and spent. This enabled them to manage disruptions due to seasonal and economic fluctuations as well as family dynamics. Examples include adjustments of the amount and deadlines of saving, taking and repaying loan or a broad understanding of health care costs. What all the groups voiced was a wish to acquire their money back at the end of the cycle. They saw this as an advantage of saving and lending rather than paying an insurance premium. They got the money they saved or paid back for a health loan at the end of the cycle and could use it to pay for other livelihood expenses or invest it in business activities. 

We further found that savings groups valued social responsibility. Being a responsible member involved regular participation in the meetings and the activities of the group as well as acting in a respectful and polite way towards the committee and the other members. Going against group norms caused conflict. In one group, for instance, a member delayed paying back her loan several times. The other members agreed to repeatedly extend the deadline because they knew her difficult family health situation. When she started wrongly accusing fellow members, the group decided to take her to the Village Executive Officer. He reminded her of her responsibility towards the group and in turn asked the group to expand the deadline one more time. Women who took the group rules seriously and thus acted responsibly could feel protected by the responsibility of the others to help them master future threats including financial health risks.

A fifth point is that savings groups provide a platform for visitors from the government or from NGOs and enable the members to meet and have a direct exchange with persons they would otherwise never have the opportunity to meet. Through the participation in such encounters, women gain confidence in dealing with powerful actors. The savings group committees consisting of a chairperson, a secretary and a treasurer took the lead in such activities, but they also encouraged and trained members to take an active role, for instance in voicing their concerns according to the formal protocol of government meetings and workshops. The leaders of one group, for example, took all the members to complain to an iCHF Enrollment Officer who did not fulfill his role properly. We also heard of group members who directly called the district iCHF coordinators, in some cases not to complain but to thank them for the good services they had received. 

Moreover, and this is the last point, participating in a savings group opens up many doors to projects, programs and grants. As a woman of the Neema group has put it: 

“Nowadays, women are in groups. So many things, loans, prizes and grants, they pass through groups. So, if I leave the group and just stay at home, will the government ever notice me? So, if a loan comes from the government, we write a letter to the government and they give us loans. If I stay at home, will I get that loan?” (Martha 28 July 2019, interview by M. Mshana and A. Orschulko).

If women comply with the rules of the group, they can access the social capital generated by the group. Groups such as Neema in Igota who know the rules of the development game attract government officials, NGOs and experts and win them as powerful allies and sponsors. These actors in turn provide access to money, material goods as well as education and thus foster the capacities of the groups and individual members. Social capital thus transforms into economic capital as well as cultural capital. 

### 3.3. Capacities to Access Cultural Capital

Bourdieu’s concept of cultural capital draws attention to how savings groups foster access to knowledge, skills and education considered as relevant and legitimate by powerful development actors. A premise held by development actors, also in the field of social health protection, is that change requires a particular mindset, including an openness for new ideas and an orientation towards the future. We found that savings groups open a space for women to develop such a mindset, as the following statements of Neema group members illustrate. 

“Before joining the group, I used to stay at home with my own ideas without sharing them with anyone. My own ideas, individually, did not give me a guide. And there were few ideas. But after joining the group, we sit down together, and we exchange ideas. That’s when I saw it really helps me to expand my understanding.” (Agnela, chairperson, interview by M. Mshana and A. Orschulko, 28 July 2019).

“Earlier, I was jobless and hopeless. But now, since I’ve joined the group, I have got a certain kind of enlightenment, and I have been awoken because of the different ideas from different people. People are offering ideas, and when you add them together, you can financially rise up.” (Agrippina, interview by M. Mshana and A. Orschulko 24 July 2019).

“Before joining this group, my mind was not really thinking any further. I was just dependent on my husband to provide assistance. This is the way people live here. But since joining the group, I have seen that my thinking has improved. I am thinking farther, and I make sure that I am taking care of my family. I involve myself in small businesses, and if I don’t have any savings in my house, I go to the group.” (Rehema, interview by M. Mshana and A. Orschulko 29 July 2019).

In addition to this general capacity to expand the mind and become more forwardlooking, savings groups enable women to develop technical knowledge and skills. Women learn to successfully apply for and pay back loans to cover health care costs. They improve their understanding of how the iCHF works, how to enrol and renew membership, and what entitlements should come with membership. Within the groups, not every member has a full understanding of these processes, but they can learn from the experiences and aspirations of others, seek their advice and ask them for guidance. 

Group discussions about positive and negative experiences gradually take the form of shared aspirations and narratives. Some groups and group members even develop skills in iCHF advocacy, both on a personal and on a community level, as the following examples from Igota show. In the first example, a woman responded to the question whether she thought her involvement in this group empowered other women to join the iCHF:

“Yes it does. When we are in the farm fields and we are just having normal conversations, we are talking about the iCHF, and sometimes we advise them. It’s like we’re doing a campaign. When you advise someone very well, then they agree and they enroll. And when they see the discomfort they have at the hospital (e.g., to pay out-of-pocket), they enroll. Sometimes, we say that they need iCHF because they are parents. We, the parents, we need to be the ones that are insisting. Especially women because as a woman, you are the one who will be struggling with a child in the middle of the night. You are tired, you have to take your child on your back, and sometimes, the fathers don’t care as much as we do. That is how we advise them.” (Fatuma, interview by M. Mshana and A. Orschulko 1 June 2019).

The second example refers to a performance with which the Igota group surprised the research team during the review meeting at the end of the first cycle. 

The plot of the 15 min performance delivered a simple and clear message: If you do not join the iCHF, you will lose money, and you may even lose your child. The women played a family of four—mother, father and two children. The daughter falls sick in the middle of the night. She has convulsions, and the parents bring her to the health dispensary. Since the family has not enrolled in the iCHF, they cannot even acquire a consultation by the nurse before paying TZS 2000 (USD 0.8). Annoyed, the father decides to consult a local healer who charges them even more and performs a ritual that does not help. The daughter dies the following night. The father regrets his decision and realizes that as an iCHF member, he would have received the right treatment for his daughter at the dispensary as part of the schemes’ benefits. (Field notes, M.M., A.O. and the research team, 25 July 2019).

During the ACCESS Project, we had seen similar performances by the Igota group and by a women group in Kisegese. The project’s social marketing team had actually scripted the story and trained the women to perform it in order to spread the CHF message within the community. The project and other development actors, including the women groups, considered this kind of training as education for community sensitization. From the perspective of social resilience, such training can be seen as an effort to improve the community capacity to access the right and legitimate knowledge of experts through a reframing of local experiences and aspirations of social health protection.

Even the moderated group discussions of the Saving4Health team can be seen as a reframing effort. It opened a space for a discussion that would otherwise not have taken place in the same form. During these discussions, the KV-HELP moderator first prompted women to exchange positive as well as negative experiences with saving, lending and the new iCHF scheme. After the peer-to-peer exchange, the moderator took up immediate concerns of the women, clarified misunderstandings, explained how the iCHF works and situated it in the broader picture of the government’s aspiration to achieve health insurance for all. Moreover, as part of the participatory research, the team encouraged women to search for potential solutions to challenges they had encountered. In some cases, the team invited iCHF enrolment officers, nurses from the village dispensary or district authorities to join the discussions. Due to the participatory orientation, the discussions served to expand the horizon of all those present. Nevertheless, the language and concepts we used, for instance Saving4Health (in Kiswahili: wekeza akiba kwa ajili ya afya) or “insurance” (in Kiswahili: bima), referred to knowledge and values of the powerful world of experts, whom we represented in these meetings. This leads us to the consideration of symbolic capital.

### 3.4. Capacities to Increase Symbolic Capital

Bourdieu’s concept of symbolic capital refers to resources rooted in honor, prestige or social recognition. We found that savings groups can mobilize, combine and transform economic, social and cultural capital and thus improve access to honor, status, recognition and reputation for the group and its members. 

In the discussions and interviews, women talked about their striving for upward mobility and development and were proud of their achievements. Within the family, their access to savings for health increased their recognition or status as the following example from the Neema group illustrates:

A woman recounted how her husband fell seriously ill and had to be brought to the hospital as an emergency. She had saved money and kept some of the cash at home. Looking back, she said: “That’s when I learnt that it is very important to have savings in the house. I became very strong. My husband needed a surgery, and he got well. When he came back, his brother who studied abroad sent money and asked: ‘How did you manage to get the treatment for him?’ I told him that it was my savings, and that I was involved in a small business of selling kanga (wrap around cloth). He was very thankful. So, I have been able to rescue my husband.” (Seraphina, interview by M. Mshana and A. Orschulko, 28 July 2019).

Within the community, the Neema group set an example of what women can achieve together. During discussions, we learnt that the women groups in Iragua and in Sofi villages had recognized the achievements of the Neema group in Igota and invited the leaders to provide advice and guidance. 

Women were proud of their groups and even competitive. A common expression was “we are on fire” (meaning “we cannot be stopped”).

The Neema secretary boasted: “I thank God for giving me the idea of joining this group. You know, I am on fire. I am very, very good… You know we are on fire. I have been very brave and determined, and the whole district knows me. I am very happy. I am very successful.” (Grace, interview by M. Mshana and A. Orschulko, 17 July 2019).

The secretary, the chairperson but also other members of the Igota group spoke about their fame or reputation. They said the grants and prizes they had won in their district had made them famous. They now even earned money if they mobilized communities for women’s rights or land rights. Other groups spoke about invitations to sing and dance at village events and wedding celebrations, and one group was invited to showcase their quality rice at the annual Nane Nane exhibition in Dar es Salaam. In other words, the combination of their ability to save, take loans and rise up financially (economic capital), of being organized and reliable in terms of social responsibility within the group and the broader community (social capital), and of having skills, knowledge and education (cultural capital) improved their capacity to gain access to honor, social recognition, and reputation (symbolic capital). 

The Neema group emphasized how the CHF training they received enabled them to do advocacy, which in turn increased their recognition as local experts. They enjoyed recounting how they mobilized people, even in other villages, and how they coached other groups to join the old CHF during the ACCESS Project. For them, the CHF was “more than a health insurance” [33]. This helps to explain why the Igota women collectively joined the improved CHF, even after the premium increased threefold in 2018, and why the group they coached in Sofi followed their example.

In their advocacy, the Igota women often pointed to privileges that came with being an iCHF member. If you showed an iCHF card at the public health facility, the staff would call you before other patients, give you better service, and hand out medications, even if they were in short supply. We could not verify, whether this was actually the case in the public health facilities nearby. In Ifakara town, the iCHF team made an effort to organize preferential treatment for members of the scheme. The important point here is that the Igota women used the capacity to access preferential treatment (also a form of symbolic capital) as an argument for making membership in the iCHF attractive. Gaining the status of preferential treatment seemed to count even more than the cost-effectiveness argument. 

However, the Igota women also experienced the danger of failed aspirations. When members were not satisfied with the services or spent money on the iCHF premium without becoming ill, they turned up at their doorsteps to complain. Before the new CHF was introduced, the Igota group had stopped doing advocacy for this scheme. Other groups reported similar experiences. Even with the improved scheme, many groups could not see how their promotion of the iCHF would gain them social recognition. This proved especially difficult in villages with no or under-staffed and poorly equipped public health facilities, which could not compete with the smart private drug shops. Since the iCHF only provided access to public health facilities, it had an image problem, and women faced difficulties in convincing others to join. 

## 4. Discussion

This study has explored women’s experiences of and aspirations for social health protection with a focus on savings groups, guided by the multi-layered social resilience perspective [9]. Our participatory qualitative research design did not aim to measure the impact of savings groups with a health component [34] but to gain new insights through integrating lived experience with theoretical and academic knowledge. The findings of the study suggest that savings groups already contribute to social health protection by fostering and developing capacities for group and individual action and have the potential for increasing mobilization and transformative change. We focus on proactive and reactive capacities under optimal conditions to show their real-life and theoretical importance for creating positive change.

With regard to pro- and reactive capacities to access economic capital, three findings are particularly relevant. First, all saving groups in this study succeeded in setting and managing a Savings Fund. The total asset they accumulated in this fund per cycle (USD 1220 for 2018/19, USD 1378 for 2019/20) is comparable with the worldwide average (USD 1200) [35]. Since women accumulate savings to later capitalize loans, we suggest considering the establishment and management of a Savings Fund as a prospective capacity to access economic capital. 

Groups commonly use only a small proportion of their savings for health care loans [24,35,36]. This was also the case in our study. For our refined analysis, however, the financial value and percentage of savings used for health care loans are less important than the specific purposes for which women have used these health care loans. All groups enabled members to use loans not just for medical expenses in a strict sense of the term, but to cover direct or indirect health care costs for themselves and for sick family members. This also helped them to pay for tests or medication not covered by the iCHF, for transport to acquire iCHF or NHIF covered treatment in a referral health facility or to pay for medicines in private retail drug shops or for services in private or faith-based health facilities. Flexibility is a hallmark of savings groups and fosters the capacities of individuals and groups to manage diverse financial health risks as well as many other risks, build assets, invest in productive activities, manage cash flows and smooth incomes [24].

A third key finding is that the savings groups in our study have mobilized women to join the iCHF. Reinforcing previous messages of the ACCES Project and KV-HELP, the Saving4Health initiative encouraged women to regularly make small contributions to the Social Fund in order to save enough money to enroll into the iCHF or to pay for the annual renewal of iCHF membership. At the end of the second saving cycle, 33 percent of the women (n = 140) were active members of the iCHF, and in five groups nearly all women had joined. The government’s target set in 2011 was to reach 30 percent CHF coverage nationally [37]. It is, of course, much more difficult to reach a 30 percent target on a national scale than in our small Saving4Health initiative. Nevertheless, our findings indicate a great iCHF mobilizing potential for savings groups, but they also underline that the capacity to access economic capital alone cannot realize this potential. 

With regard to non-financial benefits of savings groups, the literature has especially emphasized social capital [24] and sociability [18]. Our findings confirm that women groups often persist over time and draw on social relations that they built up, maintained and cultivated over a long time. In the saving groups of our study, women had already experienced how reactive and proactive capacities can generate a sense of belonging, safety, trust and protection. By referring to their group as kikundi mama (the group you can always turn to if you face problems), women articulated this experience, which for them also held a promise for the future. 

The groups in our study had formed around the aspiration of mutual support to improve livelihood and master future emergencies including illness. Despite their diverse origins (ROSCAS, self-help groups, groups to implement external interventions), the women groups shared this aspiration. This reflects the long history of mobilizing women for development in Tanzania [38], also for health development [39] and for women-centric lending models inspired by the success of the Grameen Bank [40]. 

The savings group methodology does not focus on women [24,34] but many saving and lending initiatives do, including our Saving4Health initiative. An emphasis on women’s capacities is motivated by an emancipatory aim: to enable women to gain access to realms from which a male-dominated society excludes them. As a sociological study of women microcredit groups in India has shown [41] (pp. 161–162), groups foster women’s capacities through a complex set of changes centered initially on disciplined techniques of saving and lending, which require them to participate in mandatory meetings. Not the money effects a change in women’s collective and individual capacity but the structured participation in peer groups around the group action of saving with the prospect of receiving loans conveniently. The women in our study learnt by experience that members who acted responsibly according to group rules could feel protected by the responsibility of the other members to help them master future financial health risks. Women also knew that they would acquire the money saved but not used back at the end of the cycle so that they could spend it for other livelihood expenses or invest it in business activities. They saw this as an advantage of saving and lending rather than paying an insurance premium. 

Women’s participation in savings groups has not only expanded their arena of interaction to peers but also to government authorities, representatives of local and international NGOS, or experts in various fields including health. Savings groups have fostered the courage to socialize independently with a broad range of people and to articulate and assert one’s own experiences and aspirations, both as a group and individually. Similar observations have been reported from microcredit groups in India [41]. 

Participating in a savings group opened up many doors to access benefits from government and NGO projects, programs and grants. Groups such as Neema in Igota have attracted government officials, NGOs and experts and could win them as powerful allies and sponsors. These actors in turn provided access to money, material goods as well as education and thus fostered the capacities of the groups and individual members. Referring to Bourdieu’s capital theory, this shows how social capital transforms into economic capital, and as we discuss in the next paragraphs, into cultural capital. 

Previous studies have noted that savings groups open a space for women to exchange ideas and information [34] and expand their understanding of themselves and of the wider world [41]. They subsumed these aspects under social capital. We suggest that the concept of cultural capital helps to refine the analysis of these aspects. In Bourdieu’s theory, cultural capital refers to the knowledge, skills and education considered as relevant and legitimate by powerful actors. Our study paid particular attention to how savings groups foster and develop capacities to access cultural capital deemed relevant by diverse actors in social health protection. 

The women in our study used phrases such as “exchanging ideas”, “expanding my understanding”, “enlightenment”, “being awoken” and “thinking farther” when referring to changes brought about by their participation in savings groups. This language reflects the empowerment rhetoric and emancipatory agenda of women’s rights movements [38], which go beyond participatory development [42] and microcredit programs promoting “entrepreneurship” [41]. The underlying premise is that giving women direct access to credit does not automatically transform them into “entrepreneurs”. The power of participation occurs on the level of consciousness, of social awareness, of social awakening. Participation in saving groups has given women a language to articulate experiences of growing social awareness. By using this language to talk back to us, women have demonstrated their skills in addressing development actors. 

What women have further expressed in statements about “expanding understanding” and “thinking farther” is a “capacity to aspire” [25]. As with other complex capacities such as social awareness and social awakening, the capacity to aspire grows with an exposure to a wider range of experiences and imagination of alternative ways of being and doing. The participation in savings groups has enabled women in our study to expand their minds and become more forward looking by exposure to diverse experiences and practices, such as applying for and paying back loans to cover health care costs or obtaining and renewing iCHF membership. Within the groups, not every member had a full understanding of these processes, but they could learn from the experiences and aspirations of others, seek their advice and ask for guidance. 

Group discussions about positive and negative experiences with social health protection gradually took the form of shared aspirations and narratives. Some groups and group members developed these shared aspirations and narratives into iCHF advocacy in personal and communal contexts. Telling testimonies and performing small theatres are common advocacy tools, often used in social marketing. We found that some women groups had been trained in these skills. From the perspective of social resilience, such training can be seen as an effort to improve the capacity to access the right and legitimate knowledge of experts through a reframing of local experiences and aspirations of social health protection.

The moderated group discussions of the Saving4Health team can also be interpreted as a reframing effort. Our initiative opened a space for a discussion that would otherwise not have taken place in this form. Due to the participatory orientation, the discussions served to expand the horizon of all those present. Nevertheless, the language and concepts we used, for instance Saving4Health (in Kiswahili: wekeza akiba kwa ajili ya afya) or “insurance” (in Kiswahili: bima), referred to the knowledge and values of the powerful world of experts, whom we represented in these meetings.

Symbolic capital, such as cultural capital, has been neglected in the literature on savings groups. In Bourdieu’s theory, economic, social and cultural capital can be mobilized, combined and transformed to symbolic capital, defined as resources rooted in honor, status, recognition and reputation. We found that savings groups could improve women’s access to symbolic capital, collectively and individually. Many women were proud of their achievements and gave examples of their capacity to access group savings to cover health care costs. They spoke of how they became praiseworthy in the eyes of their husbands, their extended family, their neighbors and their fellow group members. Groups that were doing particularly well were invited by other women groups to coach them and gained the recognition of government authorities as well as NGOs. They received grants and prizes that made them famous within their communities and beyond. A qualitative study of microcredit groups in India also found that women highly value the social recognition that comes with securing loans and engaging in collective action [41]. 

In our study, advocacy work, also for the CHF, increased women’s recognition as local experts. At least for one group, the iCHF was “more than a health insurance” because it had improved their capacity to access symbolic capital. In their advocacy work, they even used increased recognition as a selling argument. They emphasized that iCHF members receive preferential treatment. 

In other words, the combination of women’s increased capacity to save, take loans and rise up financially (economic capital), of being organized and reliable in terms of social responsibility within the group and the broader community (social capital), and of having skills, knowledge and education (cultural capital) improved their capacity to gain access to honor, social recognition, and reputation (symbolic capital). 

The discussion of our findings has pointed out several limitations of the study: our emphasis on proactive and reactive capacity under optimal conditions, our focus on women groups, and the reframing effect of our moderated group discussions. Moreover, our study was limited to a rural area and did not account for experiences and aspirations of saving groups in urban areas. The validity and generalizability of our findings would be increase by comparative studies on mixed and men saving groups, of mixed and specific age groups, in other rural and urban parts of Tanzania and beyond.

Microcredit as well as resilience discourses have been criticized for reproducing a neoliberal optimism [43,44]. We partly agree with this criticism. It would be naïve to assume that a simple development tool such as saving and lending will enable all people in contexts of livelihood insecurity to protect themselves against financial health risks. In line with other authors, we consider transformative change of, not adaptation to, an adverse condition [45] as the goal of resilience research and contend that Bourdieu’s capital theory offers a refined frame of analysis of what may facilitate transformative change [46]. 

Our study shows how savings groups foster and develop capacities that enable women and their families to access capitals and thus generate social health protection in “informality”. From a health financing policy perspective, the informal sector refers to the population that does not receive predictable salaries or wages [11] and cannot be taxed or forced to contribute to government regulated social health protection schemes. We concur with promoters of the VSLA model that “a greater willingness to embrace the ambiguities of informality is essential if significant deepening of the financial sector [and we would add social health protection] is to occur in poor countries” [24] (p. 11). 

The argument for an expanded concept of social health protection does not contradict the UHC aspiration for governments to offer financial protection from direct payments for health services [3]. Since the introduction of user fees in health services in low- and middle-income countries in the 1990s, the poor pay all of the financial risk associated with paying for healthcare and have to make difficult choices between health and other urgent livelihood needs. Savings groups are in many ways an NGO response to these neoliberal reforms spearheaded by the World Bank and the IMF. Our study provides further evidence of how savings groups support people in paying direct health care costs. In addition, and these are new insights, savings groups increase the capacity of people to join insurance schemes such as the iCHF by offering a mechanism for small but regular savings to pay the annual insurance premium and to access loans for direct and indirect health care costs not covered by insurance schemes.

Savings groups act as nodal points between individual women and governmental and non-governmental services and schemes. We suggest considering them as partners who jointly shape the emerging field of social health protection with government and other powerful actors rather than as community entry points for brief consultations and one-way education and sensitization for government schemes. An expanded Saving4Health initiative could develop modules for moderated discussions of the respective advantages and disadvantages of savings and insurances. Such discussions could lead to a real dialogue, tapping into the social resources of savings groups, engaging them in a critical reflection of their experiences and aspirations and creating a space for their social recognition, for instance through a prize for the best advocacy initiative. This would also entail taking local concerns seriously. For the example of the iCHF in Tanzania, this would mean, for instance, providing preferential treatment for iCHF members, linking the iCHF with private drug sellers and health facilities [47] or critically reviewing the opportunities and constraints of digital participation [48]. Increasing women’s capacity for group action to push implementers for greater accountability could also go a long way towards making government insurance schemes such as the iCHF more people-centered [49]. 

## 5. Conclusions

This social resilience study shows how women savings groups shape the evolving field of social health protection in interaction with governmental and other powerful actors and have further potential for mobilization and transformative change on the move towards UHC.

## Figures and Tables

**Table 1 diseases-10-00063-t001:** Respondents taking loans for health care costs during the second cycle (1 July 2019–30 June 2020).

Location		Respondents Taking ACCESS Loans		Respondents Taking Savings Fund Loans		Respondents Taking Social Fund Loans	
Village	Members (No.)	Total	For health care costs	Total	For health care costs	Total	For health care costs
Igota	18	14	0	4	0	1	0
Iragua	17	0	0	16	0	0	0
Kisegese	21	19	0	11	1	21	21
Mkangawalo	23	8	4	12	4	21	17
Namwawala	22	20	0	21	1	21	11
Biro	34	0	0	27	5	8	8
Iputi G.-Maya ^1^	22	2	0	14	0	n.d.	n.d.
Iputi Mwamko	23	2	0	20	0	n.d.	n.d.
Iputi Juhudi	20	0	0	16	0	n.d.	n.d.
Iputi A.-Uoga	23	6	0	19	0	n.d.	n.d.
Ketaketa	48	37	0	32	0	27	0
Ngalimila	32	30	0	29	0	22	15
Taweta	28	28	2	26	0	7	7
Kiberege	30	27	0	15	0	10	5
Mbasa	26	17	0	0	0	0	0
Ruaha	31	0	0	3	1	3	0
Minepa	16	16	0	16	9	11	4
Sofi Majiji	24	8	0	3	1	0	0
Total	458	234	6	284	22	152	88

^1^ The big Iputi Group divided into four smaller ones in January 2020.

**Table 2 diseases-10-00063-t002:** Respondents with insurances, exemptions and waivers (30 July 2020).

Village	Members (No.)	CHF Members	NHIF Members	Exemptions Waivers
Igota	18	18	1	3
Iragua	17	2	0	0
Kisegese	21	20	0	0
Mkangawalo	23	9	0	0
Namwawala	22	15	1	0
Biro	34	2	1	0
Iputi G.-Maya	22	0	0	0
Iputi Mwamko	23	2	0	0
Iputi Juhudi	20	3	0	0
Iputi A.-Uoga	23	4	0	0
Ketaketa	48	9	1	1
Ngalimila	32	1	1	0
Taweta	28	3	2	0
Kiberege	30	30	30	0
Mbasa	26	3	0	0
Ruaha	31	5	0	3
Minepa	16	2	1	0
Sofi Majiji	24	24	0	0
Total	458	152	38	7

## Data Availability

Not applicable.

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
