# Peer review of "Savings Groups for Social Health Protection: A Social Resilience Study in Rural Tanzania"

_diseases, 2022, doi:10.3390/diseases10030063_

Round 1

Reviewer 1 Report

In general I suggest to shorten the manuscript avoiding redoundancy of concepts all through the article.

  • synthesize the text by eliminating redoudant information;
  • write coincise results, as it become immediately clear what was found: it is recommended to keep critical and speculative comments only within the discussion paragraph;
  • tables have to be contestualised within the text, so reader can go directly to the information needed.

Results paragraph :

Do not repropose here the aim again (see Raws: 262,263), it is not in the relevant section to be cited.

Order the presentation of results (e.g. first women's membership and then what emerged from the first and second cycles).

Contextualize the presentation of the tables, as it must be immediately clear to what they refer to.

Avoid writing in the results paragraph reflections or critical point of views, more pertinent to the discussion paragraph, instead. In general there is confusion between what is a result and what is discussed. Results paragraph must be improved.

Propose in Conclusions and not in Results paragraph the final evidence to which the results led (e.g. Raws: 385-389)

Author Response

In general I suggest to shorten the manuscript avoiding redoundancy of concepts all through the article.

  • synthesize the text by eliminating redoudant information;
  • write coincise results, as it become immediately clear what was found: it is recommended to keep critical and speculative comments only within the discussion paragraph;
  • tables have to be contestualised within the text, so reader can go directly to the information needed.

General response: Thank you for your helpful comments. We agree that this qualitative research paper is "wordy" but the redundancy is intentional to ensure the precise use of words and concepts. See also our response to Point 4.

Point 1:  

Do not repropose here the aim again (see Raws: 262,263), it is not in the relevant section to be cited.

  • Response: We have deleted the statement about the aim in Rows 262-3.

Point 2 & 3: 

Order the presentation of results (e.g. first women's membership and then what emerged from the first and second cycles).

Contextualize the presentation of the tables, as it must be immediately clear to what they refer to.

  • Response: We cannot rearrange the presentation of our results and tables because both tables refer to cycle 2. But we are very grateful that you pointed out that references to the tables were missing in the text! We have inserted them in the corresponding paragraphs, and this will contextualize the tables as you have suggested.

Point 4: 

Avoid writing in the results paragraph reflections or critical point of views, more pertinent to the discussion paragraph, instead. In general there is confusion between what is a result and what is discussed. Results paragraph must be improved.

  • Response: We understand your concern about mixing results and reflections in the result section but this is common in qualitative writing. Insights (or "results") are not considered as facts but as interpretations.

Point 5:

Propose in Conclusions and not in Results paragraph the final evidence to which the results led (e.g. Raws: 385-389)

  • Response: We followed your recommendation and deleted part of the summary statement at the end of the section on economic capital (rows 387-389).

Reviewer 2 Report

The authors have studied that how participation in saving groups by the rural women enhanced individual capacities to access, combine and transform various capitals which are in the form of insurance.  The study done by the authors is relevant and has all the merits to be published in the journal Diseases. There are 1-2 minor mistakes and needs to be reviewed by the authors.

Author Response

Reviewer Comment:

The authors have studied that how participation in saving groups by the rural women enhanced individual capacities to access, combine and transform various capitals which are in the form of insurance.  The study done by the authors is relevant and has all the merits to be published in the journal Diseases. There are 1-2 minor mistakes and needs to be reviewed by the authors.

Response:

Thank you for your positive comments and for pointing out the mistakes. We have corrected them (lines 387, 630). 

Reviewer 3 Report

Dear author's

I was pleased to review your manuscript and the subject is interesting. I have the following comments:

Please explain the limitation of your study.

Please explain the novelty of the study.

I suggest you to add a short conclusion for your article. 

Author Response

Dear author's

I was pleased to review your manuscript and the subject is interesting. I have the following comments:

General Response: Thank you for your encouraging words and helpful comments.

Point 1: Please explain the limitation of your study.

Response to Point 1: We agree and have added a paragraph in the discussion (lines 761-767).

Point 2: Please explain the novelty of the study.

Response to Point 2: We agree and have rephrased a paragraph in the introduction (lines 79-91).

Point 3: I suggest you to add a short conclusion for your article.

Response to Point 3: We agree. We deleted the last sentence of the discussion (lines 810-813) and expanded this statement as a conclusion (lines 815-819). The conclusion matches the last sentence of the abstract.